# Preclinical Immunogenicity and Efficacy Studies for Therapeutic Vaccines for Human Papillomavirus-Type-16-Associated Cancer

**DOI:** 10.3390/vaccines12060616

**Published:** 2024-06-04

**Authors:** Mohsen Mohammadi, Amara Saha, Wynetta Giles-Davis, Zhiquan Xiang, Mikhail Novikov, Mohadeseh Hasanpourghadi, Hildegund C. J. Ertl

**Affiliations:** The Wistar Institute, Philadelphia, PA 19104, USA; mmohammadi@wistar.org (M.M.);

**Keywords:** human papillomavirus, therapeutic vaccines, CD8^+^ T cell responses, innate immunity

## Abstract

The objective of this study was to conduct preclinical immunogenicity and efficacy studies with several therapeutic vaccines for human papillomavirus (HPV)-16-associated cancers expressing the early antigens E5, E6, and E7 with or without E2. The viral oncoproteins were either expressed by themselves as fusion proteins or the fusion proteins were inserted genetically into herpes simplex virus (HSV)-1 glycoprotein D (gD) which, upon binding to the herpes virus entry mediator (HVEM), inhibits an early T cell checkpoint mediated by the B and T cell mediator (BTLA). This, in turn, lowers the threshold for T cell activation and augments and broadens CD8^+^ T cell responses to the antigens. The fusion antigens were expressed by chimpanzee adenovirus (AdC) vectors. Expression of the HPV antigens within gD was essential for vaccine immunogenicity and efficacy against challenge with TC-1 cells, which express E7 and E6 of HPV-16 but neither E5 nor E2. Unexpectedly, inclusion of E2 increased both CD8^+^ T cell responses to the other oncoproteins of HPV-16 and the effectiveness of the vaccines to cause the regression of sizable TC-1 tumors.

## 1. Introduction

Human papillomaviruses (HPVs) are small double-stranded DNA viruses that encode six early (E) and two late (L) antigens. E1–E7 regulate viral transcription and replication, cell cycling, signaling pathways and apoptosis, and modulate immune responses [1]. L1 and L2 form the viral capsid. There are more than 230 types of HPVs (https://www.hpvcenter.se/human_reference_clones, 6 March 2024); 14 are considered high risk as they may lead to cancer, while the others are low risk and cause warts on the skin and on mucosal surfaces [2]. Most individuals clear an HPV infection, but in some the virus persists. Persistent infections with any of the 14 oncogenic types of HPV, with types 16 and 18 being the most common [3], can lead to anal or genital cancers, such as cervical cancer in women, penile cancer in men, or head and neck squamous cell carcinoma (HNSCC) [4]. Preventative vaccines are available for some types of HPV [5] but remain underutilized and fail to have therapeutic benefits. Cervical cancer, the most prevailing type of HPV-associated cancer, thus remains the fourth leading cancer in females aged 15–44 worldwide with an estimated 600,000 new cases each year and 350,000 deaths, mainly in middle- and low-income countries [6].

Patients with HPV-associated cancers are primarily treated with surgery, which, in advanced cases, is combined with radiation and chemotherapy. Active immunotherapy is emerging as a promising tool to treat HPV-associated cancers [7,8,9] as the early viral antigens, especially E6 and E7, are essential for the malignant phenotype of the transformed cells by inactivating two key tumor suppressor proteins; E6 targets p53 for rapid proteasomal degradation [10], while E7 destabilizes the retinoblastoma protein (Rb). The E5 protein contributes to cell transformation by affecting cell proliferation and apoptotic pathways [11].

HPV-16 oncoprotein-expressing vaccines based on DNA [12], mRNA [13], viral vectors [14,15], proteins [16], or other platforms [17] have shown efficacy in mouse models. Treatment with checkpoint inhibitors such as antibodies to PD-1 or therapeutic vaccines also achieved tumor regression or partial responses in humans with respiratory papillomatosis due to chronic airway infection by HPV-6 or -11 [18], moderate to advanced cervical intraepithelial neoplasia, and HPV^+^ head and neck cancers [19,20]. While cell transformation is commonly initiated upon integration of the viral genome and overexpression of E6 and E7 [21], HPV oncogenesis can also be linked to episomal viral DNA and overexpression of E2, E4, and E5 with minimal expression of E6 and E7 [22,23]. Accordingly, an analysis of TILs isolated from tumors of patients with HNSCC showed higher frequencies of CD8^+^ T cells to E2 and E5 than E6 and E7. The authors concluded that therapeutic vaccines for some HPV-associated malignancies should include all four of the early antigens [24].

Preclinical and clinical studies have focused on vaccines expressing mainly E6 and E7. Here, we describe preclinical data on two types of HPV-16 vaccines that, in addition to HPV-16 E6 and E7, express E5 with or without E2. Vaccines are based on E1-deleted replication-defective chimpanzee-origin adenovirus (AdC) vectors called AdC6 or AdC68 [25]. Both vaccines express segments of oncoproteins either by themselves as fusion proteins or inserted into HSV-1 gD, which, by blocking an early T cell checkpoint mediated by interactions between the B and T cell attenuator (BTLA) on T cells and the herpes virus entry mediator (HVEM on antigen presenting cells (APCs)), augments and broadens CD8^+^ T cell responses [26]. Expression of the HPV antigens within gD markedly increased vaccine immunogenicity and efficacy. Unexpectedly, the vaccine-carrying segments of E2 were not only more immunogenic but also provided superior tumor clearance in a mouse challenge model based on tumor cells, which express E6 and E7 of HPV-16 but neither E5 nor E2.

The HPV-16 E2 protein is composed of an N-terminal transactivation domain (TAD) followed by a hinge region and a DNA binding domain (DBD) [27]. E2 is rapidly ubiquinated and degraded at a site within the TAD [28]. This, in our study, resulted in the secretion of a complex composed of the N-terminus of gD containing the HVEM binding site, which may increase the ability of gD to block the interactions between BTLA and HVEM. The secreted complex also contained the C-fragment of the digested gDE7652 fragment as well as DNA that had likely been attached to E2’s DBD. The DNA, potentially upon uptake by APCs and upon endocytosis of the HVEM-bound gDE7652-DNA complex, increased activation of the APCs, which, in turn, may have contributed to enhanced CD8^+^ T cell activation and vaccine efficacy.

## 2. Materials and Methods

### 2.1. Cell Lines

Human embryonic kidney (HEK) 293, A549, and TC-1 were grown in Dulbecco’s Modified Medium (DMEM) supplemented with fetal bovine serum (FBS) and antibiotics.

### 2.2. Mice

This research adhered to the policies and guidelines of Animal Research: Reporting of In Vivo Experiments (ARRIVE), under the approval of the Wistar Institutional Animal Care and Use Committee. Female 6-week-old, C57BL/6 mice (Jackson Laboratory, Bar Harbor, ME, USA) were housed at the Wistar Institute Animal Facility, an Association for Assessment and Accreditation of Laboratory Animal Care (AAALAC)-accredited Institution. Mice were treated according to approved protocols. Experiments were conducted with groups of 5–10 mice 2 or 3 times.

### 2.3. Recombinant Ad Vectors

Ad vectors were generated from viral molecular clones [29]. Vectors were purified by Cesium Chloride gradient centrifugation, tested for content of virus particles (vp) by spectrophotometry, and infectious units by nested polymerase chain reaction (PCR) of cells infected with serial dilutions of vector using AdC-specific primers. Genetic integrity was tested by restriction enzyme digest and gel electrophoresis of purified viral DNA.

### 2.4. pSig-His-tag-gDE7652 Plasmid

Two PCR steps were utilized to insert His tag-encoding nucleotides between the signal peptide and mature sequence of HSV-1 gD. The first PCR step added His Tag to mature gD (forward primer: GGCGGATCCCATCACCATCACCATCACCATCACCATCACGGCGGATCCGATGCCTCTCTCAAGATGGCCGACC; reverse primer: AGTAGTTCCAGCGGGGCTGC) and the PCR product was used as template DNA for the second PCR (forward primer: GCGTGGTACCTCTAGAATGGGGGGGGCTGCCGCCAGGTTGGGGGCCGTGAT TTTGTTTGTCGTCATAGTGGGCCTCCATGGGGTCCGCGGCAAATATGCCTTGGCG GGCGGATCCCATCACCATCACCAT; reverse primer: AGTAGTTCCAGCGGGGCTGC), which added a signal peptide upstream of the His-tag to create pSig-His-tag-gDE7652. The final construct was electrophoresed on a 1% agarose gel and the DNA band was extracted from the gel, digested with Xba1 and Bsp E1 (R0540S, both from BioLegend, San Diego, CA, USA), and cloned into the pShuttle vector.

### 2.5. Protein Detection in Cell Lysates

HEK 293 cells were infected with 500 vp/cell of the AdC vectors, harvested 48 h later, and lysed with RIPA buffer (Thermo Fisher Scientific, Pittsburgh, PA, USA) supplemented with protease inhibitors. Protein concentration was determined from cleared supernatants using the bicinchoninic acid (BCA) Protein Assay (Thermo Fisher Scientific, Pittsburgh, PA, USA). Protein (25 µg) was separated on 12.5% SDS-polyacrylamide gels and transferred to polyvinylidene fluoride (PVDF) membranes (Merck Millipore, Burlington, MA, USA). Membranes were blocked with 5% powder milk for 1 h at room temperature, washed with phosphate buffered saline (PBS), and incubated overnight at 4 °C with a monoclonal anti-gD antibody (clone DL6, Santa Cruz Biotechnology, Santa Cruz, CA, USA) diluted to 1:2000 in 2.5% blocking buffer. Membrane was washed with tris buffered saline (TBS) and treated with a 1:5000 dilution of horseradish peroxidase (HRP)-conjugated goat anti-mouse IgG for 1 h at room temperature. After washing, Super Signal West Pico Chemiluminescent (Thermo Fisher Scientific, Pittsburgh, PA, USA) was added to the membranes and the bands were visualized by the ChemiDoc Imaging System.

### 2.6. Protein Detection in Cell Supernatants

A549 cells were transfected with pSig-His-tag-gDE7652 and the supernatant was collected 48 h later, added to nickel (Ni)-coated beads, and incubated at room temperature for 30 min. Beads were collected with a magnet, protein was separated by gel electrophoresis and detected by staining with an HRP-labeled anti-His tag antibody (Thermo Fisher Scientific, Pittsburgh, PA, USA), and an HRP-labeled anti-gD antibody.

### 2.7. HVEM Binding of the Ad Vectors’ Transgene

HVEM fused to human IgG fragment crystallizable (Fc) (ACRO Biosystems, Newark, DE, USA) at 2 µg/mL was added to supernatants of pSig-His-tag-gDE7652 transfected A549 cells and incubated at 4 °C overnight. Pierce™ Protein A Magnetic Beads (20 µL, Thermo Fisher Scientific, Pittsburgh, PA, USA) were added, and upon a 30-min incubation, beads were collected with a magnet. Laemmli sample buffer was added, and samples were visualized by western blotting using a 1:1000 dilution of an anti-HVEM antibody (clone 122, BioLegend, San Diego, CA, USA) or a 1:2000 dilution of an anti-His-Tag antibody (clone HIS.H8, Thermo Fisher Scientific, Pittsburgh, PA, USA).

### 2.8. DNA Binding to the Ad Vectors’ Transgene Product

Antigens were precipitated from the supernatants of infected cells with an anti-gD antibody coated to Pierce™ Protein A magnetic beads. Beads were washed and treated with 250 µg/mL DNase I for 30 min at 37 °C or left untreated. Samples were electrophoresed in a 1% agarose gel for 30 min and stained with ethidium bromide followed by staining with Coomassie blue.

### 2.9. Vaccination and Challenge of Mice

AdC vectors were diluted in sterile saline to 200 µL and injected at 10^10^ vp i.m. into the hind legs of mice. TC-1 cells were washed and diluted in serum-free medium and injected at various doses subcutaneously into the right flanks of mice.

### 2.10. Isolation of Lymphocytes

Lymphocytes were isolated from blood, spleens, and tumors as described [30].

### 2.11. In Vitro Stimulation Lymphocytes

Lymphocytes were stimulated with pools of peptides or individual peptides as described [30].

### 2.12. Intracelluylar Cytokine Staining (ICS) and Analyses by Flow Cytometry

ICS was conducted as described [30] with the following antibodies: anti-CD8-APC (clone 53-6.7, BioLegend, San Diego, CA, USA), anti-CD4-BV605 (clone RM4-5, BioLegend, San Diego, CA, USA), anti-CD44-Alexa Flour 700 (clone IM7, BioLegend, San Diego, CA, USA), anti-interferon (IFN)-γ-FITC (clone, XMG1.2 BioLegend, San Diego, CA, USA), anti-perforin-PE/Dazzle 594, and anti-granzyme B (GrB)-PE/Cyanine 7.

### 2.13. Dextramer Staining

Lymphocytes were stained with a violet live/dead dye (Thermo Fisher Scientific, Pittsburgh, PA, USA), anti-CD8-APC, anti-CD44-Alexa Flour 700, anti- programmed cell death protein 1 (PD1)-BV605 (clone 29F.1A12, BioLegend, San Diego, CA, USA), anti- lymphocyte-activation gene 3 (LAG3)-BV650 (clone C9B7W) or anti-LAG-3 PE/Fire (clone NIM-R8, CA), anti- T-cell immunoglobulin and mucin-domain containing-3 (TIM3)-BV785 (clone RMT3-23), anti- T cell immunoreceptor with Ig and ITIM domains (TIGIT)-PE/Dazzle 594 (clone 1G9), anti- killer cell lectin-like receptor subfamily G member 1 (KLRG1)-PerCP/Cyanine 5.5 (clone UC10-4B9), and anti-PD1-BUV395 (clone J43BD,) anti-cytotoxic T-lymphocyte-associated protein 4 (CTLA4) (Clone:UC10-4B9), anti-KLRG1 (clone: 2F1/KLRG1) (all antibodies were from BioLegend, San Diego, CA), and a major histocompatibility complex (MHC) Dextramer H2-Db/RAHYNIVTF/APC to the immunodominant epitope of E7 (RAHYNIVTF) (Immudex, Copenhagen, Denmark). Cells were washed and analyzed by a BD FACS Celesta (BD Biosciences, San Jose, CA, USA) and BD FACSDiva software Version 9.6 (built 2022_08_29_07_20) .

### 2.14. Tumor Growth

Tumor length (L) and width (W) were measured, and the tumor sizes were computed using the formula V = (L × W × W)/2, where V is tumor volume, W is tumor width, L is tumor length.

### 2.15. Immunohistochemistry

Tumors were fixed in 10% neutral buffered formalin and embedded in paraffin blocks. Slices thereof were deparaffinized, rehydrated, mounted onto microscope slides, blocked using 2% albumin for 30 min at room temperature, and then stained with a 1:100-diluted anti-mouse CD8 (clone: 53-6.7, BioLegend, San Diego, CA, USA) and a 1:1000 diluted HRP-Anti-IgG mouse (clone: Poly4053, BioLegend, San Diego, CA, USA) followed by staining with BD Phaminogen TM DAB Substrate Kit (BD Biosciences, San Jose, CA, USA). Slides were scanned with Olympus BX53 Fluorescence Microscope BX53F-AV (Sanford, NC, USA).

### 2.16. Generation and Preparation of Bone Marrow Dendritic Cells (BMDCs)

Immature BMDCs were produced as previously described [29].

### 2.17. Stimulation of BMDCs

BMDCs grown in 6-well microplates at a density of 5 × 10^5^ cells/well were incubated with 10 µM of supernatants of AdC-infected cells purified with anti-gD antibody coated Protein A magnetic beads, 1 ug/mL lipopolysaccharide (LPS) (Sigma Aldrich, St. Louis, MO, USA), or medium. Cells were harvested after 20 hrs and stained with APC-anti-mouse CD11c (clone: N418, eBioscience, San Diego, CA, USA), Alexa Flour 700-anti-mouse CD86 (clone: GL-1, BioLegend, San Diego, CA, USA), PE/Cy5-anti-mouse CD40 (clone: 2/23, BioLegend, San Diego, CA, USA), and violet live/dead dye (Thermo Fisher Scientific, Pittsburgh, PA, USA).

## 3. Results

### 3.1. Vaccine Constructs

Vaccine inserts were designed as follows. For the E765 insert, we used most of the E7 and E6 sequences. For some experiments, we used the wild-type versions of the proteins; for others, we incorporated mutations to prevent p53 or Rb degradation and added a 72 amino acid (AA) long segment of E5. For the E7652 insert, we screened E7, E6, E5, and E2 for epitopes able to associate with multiple HLA genotypes using epitope prediction algorithms (http://tools.iedb.org/main, 6 March 2024). In addition, we screened sequences for regions of homology between different types of HPVs to potentially broaden the breadth of responses to HPV types other than 16. We selected two E7 segments that are 20 and 59 amino acids (AA) in length, respectively, omitting the part that binds Rb. For E6, we selected two fragments, 25 and 45 AA in length, and incorporated a mutation into the shorter fragment present in a strong T-cell epitope of E6 of HPV-18. For E5, we selected a 48 AA segment, and for E2, two segments, which are 92 and 39 AA in length (Appendix A). For the E765 insert, the different segments encoding the HPV oncoproteins were separated by sequences encoding glutamic acid (G) and alanine (A); for the E7652 sequence, the different cDNA fragments were separated by AA (Figure 1A). As both AAs are non-polar and neutral amino acids and thereby very similar, this change should not have affected the structure of biological functions of the fusion proteins. The sequences encoding the HPV fragments were genetically fused into HSV gD after base pair 732. For the E765 insert, we used full-length gD; for the wildtype sequences of HPV-16, we used a gD from which the transmembrane domain had been removed (AdC68-gD-TME765). For some of the protein expression analyses we developed a plasmid vector termed pSig-His-tag-gDE7652 expressing gDE7652 into which we incorporated a His-tag directly downstream of gD’s signal sequence.

To generate AdC vectors, the different sequences were inserted into the viral molecular clones of E1- and E3-deleted AdC68 or AdC6—two closely related viruses that belong to species E of adenovirideae [31]. Viruses, once rescued, were tested for protein expression in infected cells by western blot with a gD-specific antibody that binds to AAs 272–279 of gD [32], which are located behind the insertion site of the HPV insert. Constructs which carry a His-tag were also probed with an antibody to the tag. For some of the sequences, HVEM binding was confirmed by immunoprecipitation.

Lysates of AdC6-gDE765-infected cells showed binding of the gD antibody to a protein of the expected size of ~80–90 kilo Dalton (kDa) (expected size: 87.6 kDa). A similar-sized faint band was also detected in lysates of cells infected with AdC6-gDE7652 (expected size: 80.8 kDa), which showed a more prominent band with a molecular weight of ~45–50 kDa (Figure 1B). The E2 of HPV has a very short half-life due to ubiquitin-mediated degradation targeting the TAD around AA 50 towards its N terminus [28], which could fragment the transgene product into an N-terminal fragment of ~56 kDa and a C-terminal fragment of ~25 kDa. The gD antibody binds after the insertion site of the HPV sequences and can thus, in cases where the protein is digested within E2, only detect the C-terminal part of gD, suggesting that the prominent 45–50 kDa band in the AdC6-gDE7652-infected cell lysate may reflect that this E2 segment, together with the C-terminus of gD upon binding to DNA, forms multimers [33]. This was confirmed by treatment of the cell lysate with DNAse, which, by degrading accessible DNA, disrupts multimer formation and revealed a fragment of ~25 kDa, although most of the gD antibody binding protein appeared to remain in a multimeric form (Figure 1A).

To investigate secretion of the transgene product into the cell culture medium, the supernatants of cells transduced with the pHis-Tag-gDE7652 plasmid were treated with Ni-coated beads to pull down the secreted antigen with the N-terminal His tag, followed by its detection by western blot using the anti-His tag antibody (Figure 1B). To determine if the secreted form of His-Tag-gDE7652 retained its HVEM-binding ability, the supernatant of cells transduced with pSigHis-tag-gDE7652 was treated with HVEM coupled to Ig Fc (HVEM-IgG Fc); the complex was isolated with protein A-coated beads and the E7652 protein was visualized by western blot with the antibody to the His-tag. To ensure the specificity of the assay, we used two controls; one was based on protein A-coated beads treated with the supernatant but without the HVEM-IgG Fc fusion protein, and the other based on HVEM-IgG Fc mixed with protein A beads but not treated with culture supernatant. Western blot analysis with anti-His tag revealed a band of about 60 kDa (corresponding to the N-terminal segment of His-tag-gDE7652) in samples either treated with Ni-coated beads or HVEM-IgG Fc and protein A beads, but not in the control samples (Figure 1C).

We repeated the experiment and, upon purification of the supernatant with Ni-beads, again detected a ~60 k DDa band corresponding to the N-terminus of the transgene product with the antibody to the His-tag. In addition, we detected a slightly smaller band of ~50 kDa with the antibody to gD which likely reflects dimers of the C-terminal portion of the degraded gD-His-tagE7652 protein (Figure 1D). We then used beads coated with an anti-gD antibody to isolate proteins from the supernatant of AdC6-gDE7652- or AdC68-gDE765-infected cells and probed the bound components for the presence of DNA using agarose gel electrophoresis with ethidium bromide and Coomassie blue staining. The gD binding protein complexed from AdC6-gDE7652 but not AdC68-gDE765-infected cells showed a clear band of DNA that was diminished upon DNAse treatment (Figure 1E). The same band was detected upon staining for proteins.

In summary, these data indicate that the gDE7652 protein is degraded within E2. A macromolecular complex consisting of the N-terminal part and dimers of the C-terminal part bound to DNA is secreted. The complex retains the ability to bind to HVEM.

### 3.2. T Cell Responses to the Vaccine Constructs in Naïve Mice

The vaccines were tested for induction of CD8^+^ T cell to the HPV antigens expressed by the AdC vectors. To this end, groups of C57Bl/6 mice were vaccinated with 10^10^ vp of AdC6-gDE7652, AdC6-E7652, AdC68-gDE765, or a control vaccine. Splenocytes were tested one month later for CD8^+^ T cells responses upon a short stimulation with peptide pools for the HPV inserts, followed by surface staining for CD8 and CD44, and intracellular staining (ICS) for granzyme B (GrB), interferon (IFN)-g, and perforin (Figure 2A, Appendix A). Mice immunized with AdC6-gDE7652 showed significant responses to peptide pools of E2, E5, and E7, with those to E7 being most potent. The AdC6-E7652 vaccine failed to induce a significant response to any of the peptide pools while AdC68-gDE765 elicited CD8^+^ T cell responses mainly to E7. Responses to E5 and E6 were detectable but low. Responses were higher and more polyfunctional upon immunization with AdC6-gDE7652 than AdC68-gDE765, which, considering that the latter expresses longer parts of the E5, E6, and E7 inserts, was unexpected. The higher responses are unlikely to reflect the small differences in the vaccine backbones, i.e., AdC6 and AdC68, as these two vectors are closely related.

To assess the breadth of CD8^+^ T cell responses, splenocytes of mice vaccinated with AdC68-gDE765 or AdC6-gDE7652 were tested upon stimulation with individual peptides representing the HPV sequences of the two inserts. As shown in Figure 2B–E, both vaccines induced responses to multiple peptides; responses to E6 and E7 were broader upon immunization with AdC6-gDE7652, while the AdC68-gD765 vaccine stimulated a broader response to E5. The AdC68-gDE765 vaccine induced T cells that recognized 27% of the E6, 33% of the E7 peptides, and 38% of the E5 peptides; the AdC6-gDE7652 vaccine induced responses to 45% of the E6, 58% of the E7 peptides, and 14% of the E5 peptides. CD8^+^ T cells to E2 peptides were only elicited by the AdC6-gDE7652 vaccine and they recognized 67% of the peptides (Figure 2C,E). Responses to peptides whose sequence was present in both inserts were, especially for E7, higher in AdC6-gDE7652- than AdC68-gDE765-immunized mice as exemplified by peptide DEIDGPAGQAEPDRA (12.6% vs. 6.2% of CD8^+^ T cells responded).

In summary, the enhanced, broadened, and more polyfunctional CD8^+^ T cell responses to the HPV oncoproteins E5, E6, and E7 that were observed upon inclusion of the E2 fragments into the vaccine depended on co-expression of HSV-1 gD.

### 3.3. Effectiveness of Vaccines to Reduce Tumor Progression

To test the effectiveness of the vaccines in halting tumor progression, we used a therapeutic model in mice that had been challenged subcutaneously with TC-1 cells [34]. We tested the AdC68-gDE765 vaccine in comparison to a control vaccine in mice that had been injected 3 days earlier with 5 × 10^4^ (Figure 3A) or 5 × 10^5^ TC-1 cells (Figure 3B). A total of 50% of mice that received the low tumor cell dose remained tumor-free after vaccination with AdC68-gDE765 compared to 10% of the controls. Tumor progression in mice that were not protected was indistinguishable between the two groups in the low dose TC-1 challenge experiment (Figure 3A) but delayed upon challenge with the high tumor cell dose upon which all mice initially developed tumors. A total of 30% of vaccinated mice transiently became tumor-free but then, after ~2–3 weeks, relapsed (Figure 3B).

We compared AdC6-gDE7652 to AdC68-gDE765. Mice were challenged with 2 × 10^5^ TC-1 cells and vaccinated 3 days later (Figure 3C). All mice immunized with the control vaccine rapidly developed tumors, as did initially 70% of the AdC6-gDE7652-vaccinated mice and 80% of AdC68-gDE765-vaccinated mice. The AdC6-gDE7652 group very rapidly controlled the tumors and then all mice remained tumor-free. Over the course of 2 months, 40% of the AdC68-gDE765-vaccinated mice relapsed and eventually had to be euthanized.

We tested the AdC6-gDE7652 vaccine in comparison to the AdC6-E7652 or a control vaccine using an intermediate tumor cell dose of 2 × 10^5^ TC-1 cells for challenge (Figure 3D). When mice were vaccinated 3 days after tumor cell injection, all mice developed tumors. Those immunized with the AdC6-gDE7652 vaccine went, within ~2 weeks after challenge, into complete remission and remained tumor-free until the end of the observation period. AdC6-E7652- and control-vaccinated mice had rapidly progressing tumors indicating that inclusion of gD was crucial for the vaccine’s efficacy.

To increase the stringency of the experiment, mice were challenged with a lower dose of 5 × 10^4^ TC-1 cells and vaccination with AdC6-gDE7652 (*n* = 25), AdC6-E7652 (*n* = 10), or a control vaccine (*n* = 5) was delayed until day 9 after challenge (Figure 3E). All mice developed tumors that could be detected at the time of vaccination. Five mice of the two vaccine groups with tumors of a size that made complete regression unlikely were euthanized 21 days after challenge and an additional five mice were euthanized 33 (AdC68-gDE765 group) or 39 days (AdC6-gDE7652 group) after challenge to assess T cell responses (next paragraph). Tumor progression was recorded in the remaining mice. The control mice had to be euthanized within 30 days after the challenge. All AdC68-gDE765-vaccinated mice developed tumors. Including the 10 AdC6-gDE7652-vaccinated mice with progressing tumors that were euthanized for the T cell assays, 44% (11/25) of mice of this group achieved complete and sustained tumor regression (Figure 3E).

We repeated the experiment with a higher TC-1 cell challenge dose of 2 × 10^5^ cells, again delaying vaccination till day 9 after tumor cell injection. All mice had tumors at the time of vaccination, which, within 8–10 days, regressed in AdC6-gDE7652-immunized mice; they were able to control tumors for ~3 weeks, and after that, tumors started to progress. Tumor progressions in control mice and AdC6-E7652-vaccinated mice were comparable (Figure 3F).

To test if the combination of tumor cell challenge and vaccination induced a long-lasting protective response, five of the surviving mice of the AdC6-gDE7652 group were tested 20 weeks after the initial challenge for CD8^+^ T cell responses to E7. They were then, together with five control mice, injected with 10^6^ TC-1 cells and retested 5 weeks later. All vaccinated mice, unlike control mice, remained tumor-free (Figure 3G) and the vaccinated group of mice had robust and comparable E7-specific CD8^+^ T cell responses before and after challenge.

Immunohistochemical analyses of sections of tumors collected 3 weeks after TC-1 cell challenge and vaccination given 9 days later showed pronounced CD8^+^ T cell infiltrations combined with areas of necrosis in those from AdC6-gDE7652-vaccinated mice but not in those from mice that received the AdC6-E7652 vaccine (Figure 3H).

These results show that the presence of the E2 sequences within the vaccine increases its efficacy in promoting tumor regression and that gD is essential for this effect. The increased efficacy against E6^+^E7^+^TC^-^1 tumor cell challenge observed upon addition of E2 required co-expression of the oncoproteins with HSV-1 gD.

### 3.4. T Cell Responses to the Vaccine Constructs in Tumor-Bearing Mice

The finding that AdC6-gDE7652 was more effective at preventing tumor progression than AdC68-gDE765 was unexpected as the latter vaccine expresses more complete E6 and E7 sequences than the former, and T cell responses to the E2 segments, which are only elicited by AdC6-gDE7652, should not contribute to protection as E2 is not expressed by TC-1 cells. CD8^+^ T cell responses to E6 and E7 stimulated by the AdC6-gDE7652 vaccine were slightly higher, broader, and more polyfunctional than those triggered by AdC68-gDE765, but differences were overall too subtle to explain the markedly higher efficacy of AdC6-gDE7652.

We therefore tested vaccine-induced CD8^+^ T cell responses in tumor-bearing vaccinated mice. We analyzed splenocytes and tumor infiltrating lymphocytes (TILs) from mice that had been challenged with 2 × 10^5^ TC-1 cells 9 days prior to vaccination with AdC68-gDE765 or AdC6-gDE7652. T cells were tested 4 weeks after the challenge. In both vaccine groups, frequencies of CD8^+^ T cells that were stained with an MHC class I dextramer specific for the T cell receptor (TCR) to the immunodominant epitope of E7 were significantly higher in tumors than spleens, and in both compartments, they were higher in AdC6-gDE7652- than AdC68-gDE765-immunized mice (Figure 4A). A functional analysis by ICS showed higher frequencies of IFN-γ producing CD8^+^ T cells in tumors of AdC6-gDE7652-immune mice, while frequencies of CD8^+^ T cells positive for GrB, which was found in many CD8^+^ TILs even without further in vitro stimulation, were higher for TILs of the AdC68-gDE765 group (Figure 4B). Boolean gating for functions showed higher frequencies for most combinations of functions in tumors compared to spleens. Comparing TILs of the two vaccine groups showed higher frequencies of T cells producing IFN-γ with or without perforin in TILs from AdC6-gDE7652-immune mice, while those from AdC68-gDE765-vaccinated mice had higher frequencies of CD8^+^ T cells producing perforin and GrB (Figure 4C).

An analysis of the expression of differentiation markers on E7-detramer^+^ CD8^+^ T cells showed higher frequencies of E7-specific CD8^+^ T cells expressing killer cell lectin-like receptor subfamily G member 1 (KLRG1), a terminal differentiation marker, or the exhaustion markers programmed cell death protein (PD)-1, lymphocyte-activation gene (LAG)3, T cell immunoreceptor with Ig and ITIM domains (TIGIT), and T-cell Immunoglobulin domain and mucin domain 3 (TIM3) [35] on cells from tumors compared to those from spleens, but only KLRG1 and TIGIT showed differences between TILs from the two vaccine groups with higher frequencies on those from AdC6-gDE7652-immunized mice. Levels of expression of the different markers determined by the mean fluorescent intensity (MFI) of the stains again were higher on E7-specific CD8^+^ TILs than splenocytes, and for LAG3, higher on those from AdC6-gDE7652- than AdC68-gDE765-immune mice. Combinations of markers, as assessed by Boolean gating, showed significant differences for several combinations with cells positive for KLRG1, LAG3, PD-1, TIGIT, and TIM3 being more common on TILs from AdC6-gDE7652-immune mice, while those expressing LAG3, PD-1, and TIM3 were more frequent in AdC68-gDE765-immune mice (Figure 4F and Appendix A).

Another significant difference was seen for the expression levels of the E7-specific TCR determined by the intensity of the E7-dextramer stain; expression levels were significantly higher on cells from spleens and tumors of AdC6-gDE7652- than AdC68-gDE765-immunized mice (Figure 4G,H). We further analyzed the dextramer stain on E7-specific CD8^+^ TILs that expressed a different combination of markers, selecting only those populations for which enough cells could be analyzed. As shown in Appendix A, AdC6-gDE7652-induced CD8^+^ T cells carrying multiple markers, including LAG-3, expressed significantly higher levels of the dextramer-binding TCR compared to the matching AdC68-gDE765-induced CD8^+^ T cells. In general, levels were lower on CD8^+^ T cells that expressed only one of the markers or TIM3, with PD1 potentially suggesting that higher levels of TCR expression contributes to T cell exhaustion characterized by expression of multiple immune checkpoints.

We then compared T cell responses in tumor-bearing mice vaccinated with AdC6-E7652 or AdC6-gDE7652 to gain further insight in the effects of gD combined with the E2 sequences on T cell frequencies, functions, and expression of exhaustion/differentiation markers. Mice were challenged with 2 × 10^5^ TC-1 tumor cells and vaccinated 9 days later. By day 21 after challenge (day 12 after vaccination), five mice in each group that had developed small, similar-sized tumors were euthanized. Another five mice were analyzed in the AdC6-E7652 group by day 33 after challenge, and in the AdC6-gDE7652 group, which showed a marked delay in tumor progression, by day 39 after challenge, the tumors had reached sizes like those isolated 6 days early from the AdC6-E7652 group. Splenocytes and TILs were isolated and tested by flow cytometry. At both the early and late time point, spleens and tumors of AdC6-gDE7652-immunized mice showed a higher degree of CD8^+^ T cell infiltration (Figure 5A). At the early time point, E7-specific CD8^+^ T cells identified by staining with a dextramer to the immunodominant epitope of E7 were higher in AdC6-gDE7652- than AdC6-E7652-immune mice; by the later time point, frequencies increased in tumors of AdC6-E7652 mice but decreased in those from AdC6-gDE7652-immune mice (Figure 5B).

The analyses of differentiation/exhaustion markers (Figure 5C,D) showed that percentages of E7-specific TILs expressing CTLA-4 or KLRG1 were higher in spleens than tumors. In spleens, they were higher on E7-specific CD8^+^ T cells from AdC6-E7652-immune mice. In both spleens and tumors, percentages of KLRG1^+^ E7-specific CD8^+^ T cells significantly decreased by day 39 in AdC6-gDE7652-immunized animals. Frequencies of LAG3^+^ E7-specific CD8^+^ T cells declined in spleens of both vaccine groups and in tumors of the AdC6-gDE7652 group over time but remained stable in the AdC6-E765 group. PD-1 was only expressed on a few percentages of E7-specific splenic CD8^+^ T cells and was more common on E7-specific CD8^+^ TILs for which those of the AdC6-E7652 group showed a significant increase between the early and late time point. Percentages of TIGIT^+^ splenic and tumor-derived CD8^+^ T cells were comparable for the different vaccine groups and the different time points. TIM3, like LAG3, was initially high on E7-specific CD8^+^ T cells from spleens and then declined, while percentages of TIM3^+^ cells were initially higher on AdC6-gDE7652-induced TILs but then increased on AdC6-E7652-stimulated T cells and became comparable for the two vaccine groups at the later time point. The analyses of different combinations of markers showed several significant differences with increasing frequencies of specific CD8^+^ T cells expressing multiple markers over time in the AdC6-E7652 group, while the reverse trend was seen in the AdC6-gDE7652-immune group (Figure 5E,F, Appendix A).

At the early time point, the intensity of the dextramer stain was higher on CD8^+^ T cells from AdC6-gDE7652-immunized mice, and this reached significance for both splenocytes and TILs, with the latter being higher than the former (Figure 5G). Analyzing the dextramer stain on E7-specific CD8^+^ T cells, which differed in expression of exhaustion/differentiation markers, showed, for the early day-21 tumors, higher levels of dextramer staining for cells induced by AdC6-gDE7652 that expressed multiple markers. Again, there was a very pronounced reduction in dextramer staining on cells which carried either none or one or two of the tested markers. These differences were not seen for CD8^+^ T cells from tumors harvested on days 33 or 39 (Appendix A).

Functions were tested in a separate experiment in which mice were vaccinated 9 days after TC-1 cell challenge and then tested 10 days later for frequencies of E7-specific CD8^+^ T cells producing GrB, IFN-γ, perforin, or combinations thereof. In spleens, AdC6-gDE7652 induced high frequencies of CD8^+^ T cells producing all three factors or IFN-γ with or without GrB; responses to AdC6-E7652 were marginal. The control vaccine (an AdC6 vector expressing gD with melanoma epitopes) failed to induce detectable frequencies of E7-specific CD8^+^ T cells. In tumors, AdC6-gDE7652-induced E7-specific CD8^+^ T cells produced mainly all three functions or IFN-γ with GrB. Frequencies of E7-specific CD8^+^ T cells induced by AdC6-E7652 were lower and most of them were monofunctional and produced only IFN-γ. There were some background responses in tumors of mice that had been immunized with the control vaccine, presumably due to activation of T cells by antigens released from the tumor cells (Figure 5G).

In mice that were vaccinated after tumor cell challenge, the AdC6-gDE7652 vaccine induced higher and more polyfunctional CD8^+^ T cell responses than the AdC68-gDE765 vaccine. Although differences reached statistical significance, they were overall subtle. One of the most surprising findings was that CD8^+^ T cells to the immunodominant epitope of E7 showed significantly higher levels of expression of the TCR in AdC6-gDE7652- than AdC68-gDE765-vaccinated mice. Differences between CD8^+^ T cells from mice vaccinated with AdC6-E7652 and AdC6-gDE7652 were more pronounced and showed again that gD was important to elicit a strong and effective CD8+ T cell response against the HPV antigens.

### 3.5. Effects of Secreted Vaccine Inserts on Antigen-Presenting Cells

As E2 is rapidly digested, resulting in an N-terminal segment that within a gD fusion protein is secreted, we assessed if secretion of part of a gD-containing antigen in itself would enhance immunogenicity. We constructed an AdC68 vector that encodes E765 within a form of gD from which the transmembrane encoding sequences (AGAVGGSLLAALVICGIVYWMHRR) had been removed (AdC68-gD_-TM_E765) and tested immune responses to this vector in comparison to AdC68-gDE765. We would like to point out that the inserts encoded by the two vectors were, but for a modification of the Rb binding site within E7 and some minor modifications with E6, to disrupt p53 binding identical. Depletion of the transmembrane domain resulted in secretion of the gD fusion protein (Figure 6A). Nevertheless, this failed to augment E7-specific CD8^+^ T cell responses (Figure 6B) nor did it affect the intensity of the dextramer stain (Figure 6C). These results were further confirmed by an AdC vector expressing the nucleoprotein of influenza A virus within full-length or transmembrane deleted gD.

To test if the secreted vaccine inserts could contribute to the activation of innate immunity, we treated in vitro matured bone BMDCs with gD antibody purified supernatants from cells infected with AdC6-gDE7652 or, as a control, AdC68-gD_-TM_E765. LPS served as a positive control and medium as a negative control. Twenty hours after treatment, BMDCs were stained with a live cell stain and antibodies to CD11c, CD40, and CD86. As shown in Figure 6D, percentages of BMDCs positive for these two activation markers, as well as the MFI of the stains (Figure 6E), increased after treatment with LPS or the AdC6-gDE7652 supernatant but not with the AdC68-gD-TME765 supernatant, showing that E2, and potentially the DNA bound to E2, contributes to the activation of professional APCs.

## 4. Discussion

Therapeutic vaccines for HPV oncoproteins based on numerous different platforms have achieved tumor remission in preclinical models [36,37,38,39,40]. DNA vaccines expressing E7 as a fusion protein within gD have been tested repeatedly and they were efficacious if given repeatedly or together with an adjuvant or a chemotherapeutic agent [41,42,43,44]. mRNA vaccines have also showed efficacy [45], especially if they expressed a gDE7 fusion protein [46]. Although mRNA vaccines are versatile and induce robust immune responses as demonstrated during the COVID-19 pandemic [47,48], they have some disadvantages. Due to their inflammatory potential, mRNA vaccines are reactogenic and can cause severe side effects [49,50]. mRNA vaccines are thermolabile and require storage at sub-zero temperatures [51]. In addition, our knowledge of potential long-term risks of mRNA vaccines [52], including those associated with ribosomal frameshifting of the mRNA sequence, remains limited [53].

Our focus has been to develop AdC vectors [54,55]. Here, we describe preclinical studies conducted with therapeutic vaccines directed against early antigens of HPV-16. The vaccines either supply segments of E5, E6, and E7 or E2, E5, E6, and E7, which are expressed as fusion proteins within HSV-1 gD. Sequences with E6 and E7 that may contribute to transformation were removed; a sequence within E5 that is only carried by oncogenic types of HPV [56] is present in the gDE7652 but not the gDE765 sequence and may have to be modified prior to use in humans. As shown previously [26,30,41,54] and confirmed in this study, gD, which inhibits an early T cell checkpoint, increases and broadens CD8^+^ T cell responses. CD8^+^ T cell responses to the HPV antigen fragments were low or undetectable in mice immunized with the Ad vector expressing E7652 without gD, which may reflect that the selected sequences were suboptimal or, more likely, that HPV oncoproteins tend to be poorly immunogenic in mice. Unexpectedly, presence of the E2 segments not only elicited responses to E2 epitopes but also enhanced responses to other viral oncoproteins, i.e., E5 and E7. This effect depended on the presence of E2 within gD. E2 had only minor effects on the expression of differentiation and exhaustion markers. Inclusion of gD into the vaccine antigen had a more pronounced effect on CD8^+^ T cell differentiation with higher expression of exhaustion markers on CD8^+^ T cells induced by the gD-containing vaccine early during the response, while CD8^+^ T cells induced without gD showed increases in the expression of exhaustion markers over time indicating, as we previously showed [30], that gD may delay CD8^+^ T cell exhaustion.

Another effect of gD alone or in combination with E2 was that it caused a significant increase in the expression of the specific TCR, as shown by more intense staining with the E7-TCR-specific dextramer. This may render CD8^+^ T cells more efficient at responding to low-abundant MHC-antigen complexes but may also increase their susceptibility to exhaustion. The TCR is composed of the MHC class I-peptide binding and ß chains and three CD3 dimers, i.e., CD3δε, γε, and ζζ. Most of the TCR components are produced in excess and stored in the ER, except for CD3z, which is rate-limiting and allows for final assembly of the TCR complex and its transport to the cell surface [57]. TCR complexes, upon binding to their cognate antigens, cluster to increase avidity and then are rapidly internalized through clathrin-mediated endocytosis [58]. Increased TCR signaling by gD-mediated inhibition of the BTLA-HVEM pathway may reduce TCR clustering and the number of TCR-MHC-peptide interactions needed for T cells’ activation and thereby lessen TCR endocytosis, which would increase surface expression. This, in turn, could enhance and prolong effector cell responses. Interestingly, the intensity of the stain was linked to the expression of exhaustion markers, most notably LAG-3, which, upon binding to TCR-CD3 complexes, inhibits downstream signaling [59]. The finding that expression of immunoinhibitors also affected TCR density could again reflect reduced internalization.

Most remarkably, inclusion of E2 increased the vaccine’s efficacy and allowed for complete remission of tumors in all mice vaccinated 3 days after challenge with 2 × 10^5^ TC-1 cells, while under the same conditions 60% of AdC6-gDE765-immunized mice showed tumor progression. Expression of the HPV oncoproteins within gD was essential for the AdC6-gDE7652 vaccine’s efficacy, which, even if given 9 days after injection of 5 × 10^4^ TC-1, cured 40% of the mice while none survived after Ad-E7652 vaccination. TC-1 cells are derived from mouse lung endothelial cells that were transformed by HPV-16 E and E7 and c-Ha-ras [34]. The cell line fails to express E2 and CD8^+^ T cells to this antigen and thus cannot contribute to the eradication of tumor cells, indicating that the enhanced vaccine immunogenicity and efficacy upon inclusion of E2 reflected an indirect effect on other mechanisms that promote immune activation.

We assessed two potential mechanisms. First, rapid digestion of the E2 protein within the TAD results in secretion of the N-terminal fragment of the gDE7652 transgene product, which retains the ability to bind to HVEM. Nevertheless, secretion alone did not amplify the vaccine’s immunogenicity as removal of the gD’s transmembrane domain failed to increase CD8^+^ T cell responses to a gD fusion antigen. The secreted gDE7652 transgene product is composed of the N-terminal and the C-terminal fragments of the protein bound to DNA. Binding of the complex to HVEM likely causes receptor-mediated endocytosis, which, in the presence of E2, results in activation of BMDCs, as shown by an increased expression of the activation markers CD40 and CD86. As this activation was not achieved by transgenes that lacked E2, and thereby DNA, we assume that the DNA, once it was transported into the endosomes, triggered innate sensors [60]. This would open the possibility that nucleic acid bind vaccine components that can enter APCs in general augment adaptive responses upon enhancing innate immunity—a mechanism that has been suggested for the adjuvant activity of alum [61].

In summary, here we show data on two AdC-based therapeutic HPV vaccines that, upon expressing the oncoproteins fused into gD, were highly immunogenic and effective at causing tumor regression. Although clinical data of gD as a vaccine additive are not yet available, its effects on human T cell responses are being explored in an ongoing clinical trial that is being conducted in patients with chronic hepatitis B virus infections (https://clinicaltrials.gov/search?intr=VRON-0200, 6 March 2024). We also provide evidence that E2 sequences present within a vaccine, together with gD, can enhance and broaden the antigen-specific CD8^+^ T cells’ responses. Although the mechanism by which E2 enhances immunity, and thereby efficacy, remains obscure and could reflect several mechanisms, such as improved antigen presentation due to enhanced degradation of complex multimeric antigens containing E2, we favor the explanation that the secretion of an HVEM binding protein–DNA complex may augment innate, and thereby adaptive, immune responses.

## Figures and Tables

**Figure 1 vaccines-12-00616-f001:**
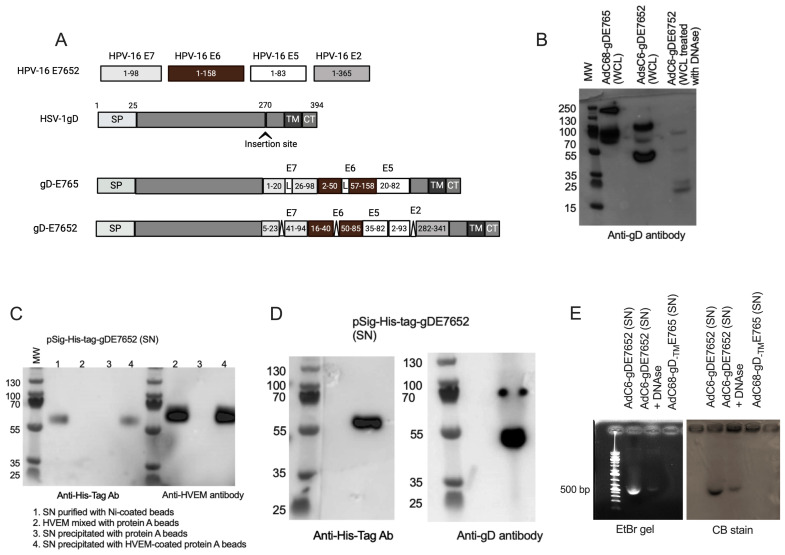
Antigens and protein expression. (**A**) Schematic diagrams of the antigens. Top shows the wild-type forms of the oncoproteins; numbers indicate the 1st and last AAs. The 2nd row shows HSV-1 gD, numbers of AA for the different domains are shown on top, SP—signal peptide, TM—transmembrane domain, CM—cytoplasmic domain. The 3rd row shows the gDE765 insert with first and last AA of each segment shown in the boxes. L—linker. The bottom row shows the gD-E7652 insert with first and last AA of each segment shown in the boxes. ∆—deletion. (**B**) Western blot of whole cell lysates (WCL) of HEK-293 cells infected with AdC68-gDE765, AdC68-gDE7652, and AdC6-gDE7652 treated with DNAse. WCLs were probed with an antibody to gD. (**C**) HEK-293 cells were transfected with pSig-His-Tag-gDE7652 (2 µg/10^6^ cells). Supernatants were harvested 48 h later and immunoprecipitated with Ni-coated beads to pull down secreted antigen harboring an N-terminal His tag (1). To show interactions of antigen encoded by pSig-His-Tag-gDE7652 that was secreted into the supernatant with HVEM, the supernatant was incubated with HVEM-IgG Fc, and then Protein A-coated beads were added to pull down HVEM-IgG Fc and its attached antigen (4). HVEM-IgG Fc incubated with Protein A-coated beads (2) or supernatant incubated with Protein A-coated beads were used as controls (3). The precipitated proteins were subjected to gel electrophoresis and visualized with an antibody to the His-tag or an antibody to HVEM. (**D**) Western blots of supernatants (SNs) of cells transfected for 48 h with pSig-His-Tag-gDE7652 and then precipitated with Ni-beads. The immunoprecipitated samples were probed with antibodies to the His-tag or gD. (**E**) SNs of AdC6-gDE7652- or AdC68-gDE765-infected cells upon precipitation with beads coated with an anti-gD antibody were either left untreated or the AdC6-gDE7652 sample was treated with DNAs. Samples were electrophoresed on 1% agarose gel, and the gel was stained with ethidium bromide (EtBr).

**Figure 2 vaccines-12-00616-f002:**
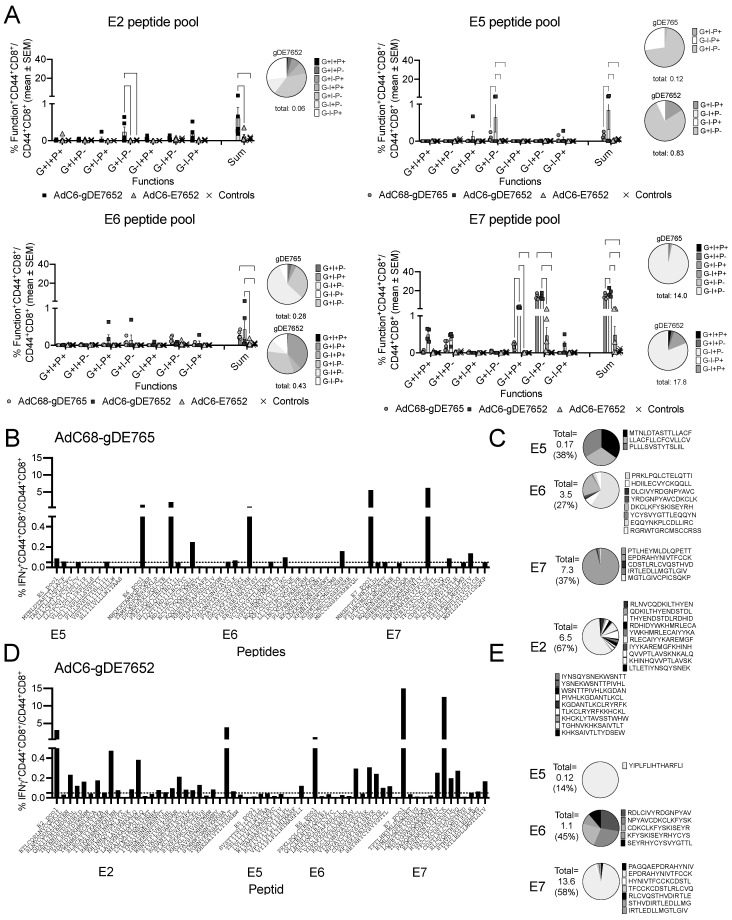
CD8^+^ T cell responses to the vaccine inserts. (**A**) Groups of 5 C57Bl/6 mice were vaccinated with 10^10^ vp of the indicated vaccines. Splenocytes were tested one month later for production of GrB (G), IFN-γ (I), and perforin (P) by ICS upon stimulation with peptide pools representing the sequences of E2, E5, E6, and E7 present in the vaccine. Naïve mice were used as controls. The bar graphs show frequencies of CD44^+^CD8^+^ T cells positive for the indicated combinations of factors determined by Boolean gating as well as sums of responses as means ± SEMs. The circle next to the bar graphs displayed for responses to AdC68-gDE765 and AdC6-gDE7652 (responses to AdC6-E7652 were too low) shows the distribution of the different combinations of functions. (**B**–**E**) Splenocytes of AdC68-gDE765- and AdC6-gDE7652-immune mice were tested by ICS for IFN-γ production in response to peptide pools as in (**A**) and individual peptides representing the sequences of the oncoprotein fragments. Magnitude of responses are shown for AdC68-gDE765 (**B**) and for AdC6-gDE7652 (**D**). Circles show relative portion of CD8^+^ T cell responding to the different peptides after vaccination with AdC68-gDE765 (**C**) or AdC6-gDE7652 (**E**).

**Figure 3 vaccines-12-00616-f003:**
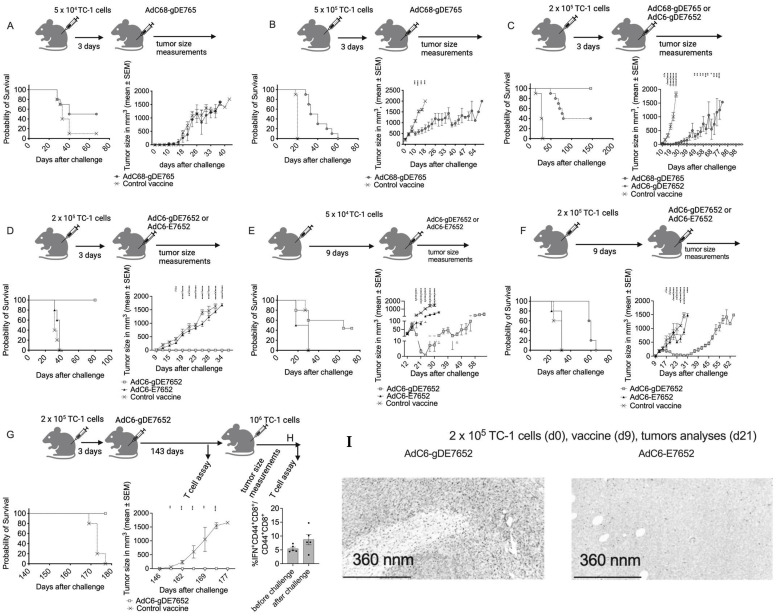
Effect of therapeutic vaccination on tumor progression. (**A**) Mice were challenged with 5 × 10^4^ TC-1 cells and vaccinated 3 days later with 10^10^ vp of AdC68-gDE765 or a control vaccine (either AdC6-gDMelapoly or AdC6-gDgag). Tumor progression was monitored, and mice were euthanized in this and the other experiments once tumors reached a volume ~1.5–2cm^3^. The right graph shows survival curves, which were not significantly different between the two groups. The left graph shows tumor progression in mice that developed tumors. Significant differences by multiple *t*-test are shown as stars above the lines. In this and all following graphs, numbers of stars indicate range of *p*-values: (^-^) *p*-value > 0.05, (*) *p* value 0.01–0.05, (**) *p*-value 0.001–0.01, (***) *p*-value 0.0001–0.001, (****) *p* value < 0.0001. (**B**) Mice were challenged with 5 × 10^5^ TC-1 cells and vaccinated 3 days later with 10^10^ vp of AdC68-gDE765 or a control vaccine. Survival on the left was significantly different between the two groups (*p* > 0.0001 by log-rank test) and tumor progression was significantly delayed in AdC68-gDE765-vaccinated mice (2-way ANOVA). (**C**) Mice were immunized with 10^10^ vp of AdC68-gDE765, AdC6-gDE7652, or a control vaccine 3 days after challenge with 2 × 10^5^ TC-1 cells. Both vaccines significantly increased survival (*p* < 0.0001), and survival was also significantly different between AdC68-gDE765- and AdC6-gDE7652-vaccinated mice (*p* = 0.004). AdC68-gDE765-immunized mice showed a significant delay in tumor progression compared to control mice. The following comparisons by 2-way ANOVA for the early time points (d12-27) are shown as in (**C**): top is AdC6-gDE7652 to control; middle is AdC68-gDE765 to control; bottom is AdC6-gDE7652 to AdC68-E765. Later time points were analyzed by multiple *t*-test for the two remaining groups. (**D**) Mice were challenged with 2 × 10^5^ TC-1 cells and vaccinated 3 days later with 10^10^ vp of AdC6-E7652, AdC6-gDE7652, or a control vaccine. Both survival curves and tumor progression showed significant differences between AdC6-gDE7652-vaccinated mice and AdC6-E7652- or control-vaccinated mice (*p* < 0.0001 by log-rank test for survival and for tumor progression by 2-way ANOVA [early time points] and multiple *t*-tests [late time points]). The following comparisons are shown by stars above each time point: top is AdC6-gDE7652 to control; middle is AdC6-E7652 to control; bottom is AdC6-gDE7652 to AdC6-E7652. (**E**) Mice were challenged with 5 × 10^4^ TC-1 cells and vaccinated 9 days later with AdC6-E7652, AdC6-gDE7652, or a control vaccine. AdC6-E7652 did not increase survival, unlike AdC6-gDE7652, which showed a significant increase in survival compared to AdC6-E7652 (*p* = 0.0011) and the control vaccine (*p* = 0.039). The delay in tumor progression comparing vaccinated mice to control mice by multiple *t*-tests was only significant for AdC6-gDE7652. (**F**) Mice were challenged with 2 × 10^5^ TC-1 cells and vaccinated 9 days later with AdC6-E7652, AdC6-gDE7652, or a control vaccine. Differences in survival curves between AdC6-gDE7652-vaccinated mice and controls (*p* = 0.0031) or AdC6-E7652-vaccinated animals (*p* = 0.0035) were significant. The results of the 2-way ANOVA analysis are shown as in (**C**). (**G**,**H**) Mice that upon challenge with TC-1 cells and vaccination with AdC6-gDE7652 failed to develop progressing tumors (*n* = 5), together with 5 naïve control mice, were bled and their PBMCs were tested by ICS for production of IFN-γ in response to the E7 peptide pool. Mice were then, 146 days after the initial challenge, injected with 1 × 10^6^ TC-1 cells. Control mice rapidly developed tumors while the vaccinated survivors remained tumor-free (**G**). PBMCs of vaccinated mice were tested 7 days after rechallenge for E7-specific CD8^+^ T cell responses by ICS for IFN-γ (**H**). (**I**) Tumor section from TC-1-challenged mice vaccinated with AdC6-gDE7652 or AdC6-E7652 stained with an antibody to mouse CD8.

**Figure 4 vaccines-12-00616-f004:**
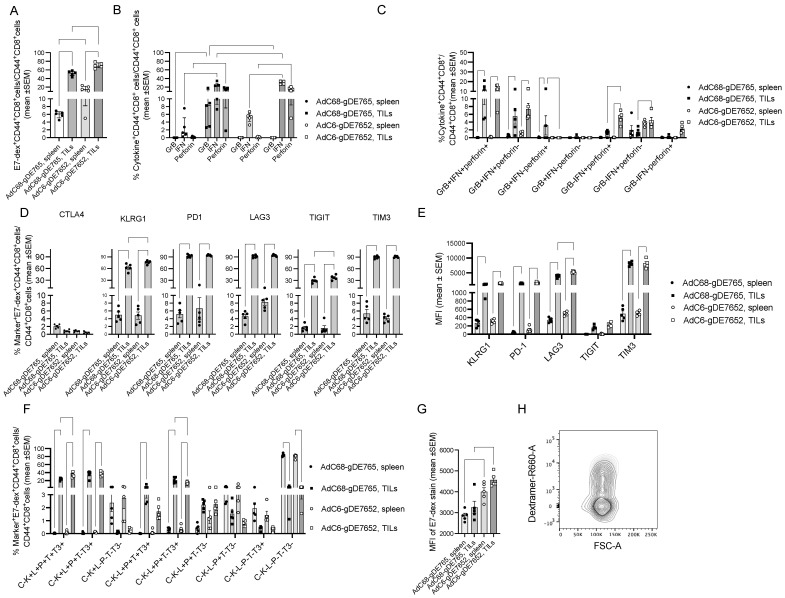
Comparing CD8^+^ T cell responses to AdC68-gDE765 and AdC6-gDE7652 in tumor-bearing mice. (**A**) Frequencies of T cells that were stained with an E7-specific dextramer and antibodies to CD8 and CD44 over all CD44^+^CD8^+^ cells are shown in this and subsequent graphs for individual mice with bars indicating means ± SEM. Significant differences by 2-way Anova in this and subsequent graphs are indicated by lines with stars above as in Figure 2. (**B**) shows frequencies of CD44^+^CD8^+^ cells over all CD44^+^CD8^+^ cells producing GrB, IFN-γ, or perforin in response to a short stimulation with E7 peptides. (**C**) Frequencies of E7-specific CD8^+^ T cells producing combinations of factors. (**D**) Expression of the indicated markers on E7-dextramer^+^CD44^+^CD8^+^ cells. (**E**) Mean fluorescent intensity (MFI) of the indicated markers on E7-dextramer^+^CD44^+^CD8^+^ cells. (**F**) Combinations of exhaustion/differentiation markers on E7-dextramer^+^CD44^+^CD8^+^ cells. Only those combinations that showed marked differences between the groups are shown. The full data set is shown in Appendix A. (**G**) MFI of the E7-specific dextramer stain. (**H**) Flow contour blot for the dextramer stain over forward scatter area (FSC-A) comparing representative E7-dextramer^+^CD8^+^ samples of mice immunized with AdC6-gDE7652 (light grey) or AdC68-gDE765 (dark grey).

**Figure 5 vaccines-12-00616-f005:**
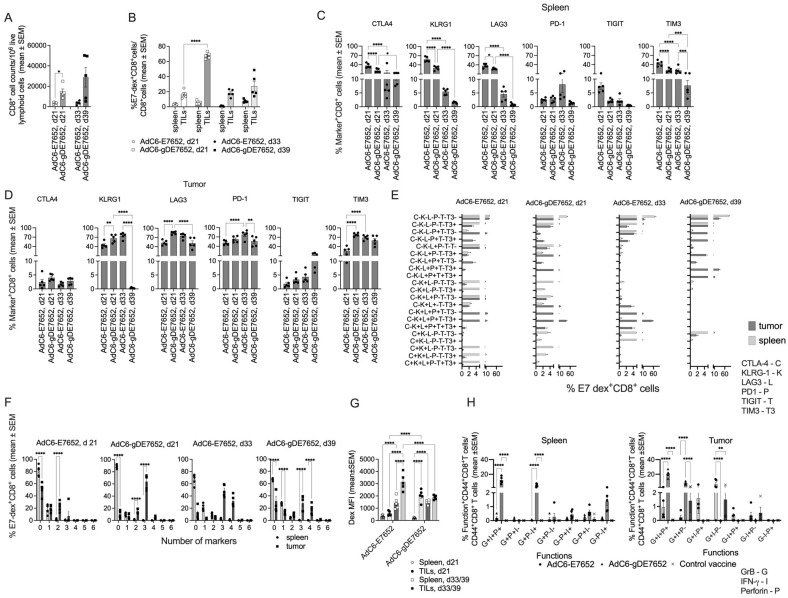
Comparing CD8^+^ T cell responses to AdC6-E7652 and AdC6-gDE7652 in tumor-bearing mice. Mice were challenged with 2 × 10^5^ TC-1 tumor cells and vaccinated 9 days later. Splenocytes and TILs were analyzed 21 and 33 or 39 days later. (**A**) Numbers of CD8^+^ cells in tumors normalized to 10^6^ live lymphoid cells. (**B**) Frequencies of E7-dextramer^+^CD8^+^ T cells in spleens and tumors. (**C**,**D**) Percentages of E7 dextramer^+^CD8^+^ cells from spleen (**C**) or tumors (**D**) expressing the indicated markers. (**E**,**F**) Percentages of E7-dextramer^+^CD8^+^ cells from spleen or tumors expressing the indicated combination of markers (**E**) or the indicated numbers of markers (**F**). (**G**) Intensity of the dextramer stain. (**H**) Percentages of CD8^+^ splenocytes or TILs producing the indicated combinations of factors. Data were compared by 2-way ANOVA and significant differences are highlighted as described in legend of Figure 2.

**Figure 6 vaccines-12-00616-f006:**
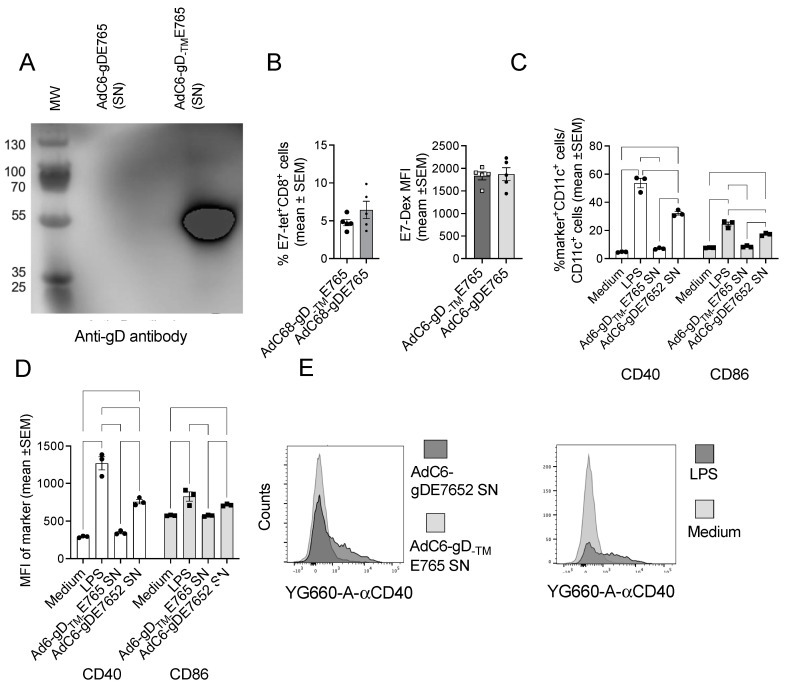
Compare the immunogensity of vaccine expressing gD or gD lacked TM. (**A**) Western blot of SNs of cells infected with AdC6-gDE765 or AdC6-gD_-TM_E765. (**B**) Frequencies of E7-tetramer^+^ CD8^+^ T cells in blood of mice injected 2 weeks previously with the indicated vectors at 10^10^ vp. (**C**) MFI of the dextramer stain. (**D**) Stimulation of BMDCs with secreted forms of gD-E7652 or gD-E765. The secreted antigens used for treatment of BMDCs were obtained from cell cultures infected with AdC6-gD-E7652 or AdC6-gD_-TM_E765. BMDCs treated with LPS or PBS served as positive and negative controls, respectively. Stimulation of BMDCs was assessed by testing for expression of CD40 and CD86. (**C**) Shows percentages of cells positive for these markers, (**D**) shows levels of expression, (**E**) shows representative flow histograms for BMDCs stained with an antibody to CD40.

## Data Availability

All data, code, and materials used in the analysis are available upon request to any researcher for purposes of reproducing or extending the analysis. Transfer of some of the material may require a transfer agreement between the institutions. All data but for one confirmatory study listed are available in the main text or the Appendix A.

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
