# Peer review of "Preclinical Immunogenicity and Efficacy Studies for Therapeutic Vaccines for Human Papillomavirus-Type-16-Associated Cancer"

_vaccines, 2024, doi:10.3390/vaccines12060616_

Round 1
Reviewer 1 Report
Comments and Suggestions for Authors
The study developed two therapeutic vaccines targeting HPV-16 based on the AdC vector, one containing the full sequences of E5, E6, and E7, and the other containing different segments of E2, E5, E6, and E7. These vaccines, expressed by fusion to gD, demonstrated good immunogenicity and showed some efficacy in mice challenged with TC-1 cells. Particularly, the vaccine containing E2, when co-expressed with gD, enhanced and broadened the specific CD8+ T cell responses to antigens. It not only elicited responses to E2 epitopes but also augmented responses to other viral oncoproteins such as E5 and E7. However, the mechanism by which E2 enhances immunity and its efficacy remains unclear. This study has some innovative aspects, especially in the preliminary exploration of the anti-tumor effects of HPV early protein E2 and its potential mechanisms of action. My comments are as followed:
Major Comments
1. Several experiment vaccines were constructed in present study. It is recommended to include a schematic diagram in the results section, clearly illustrating the amino acid sequences of each HPV antigen and the gD protein encoded by the adenovirus vector. This diagram should demonstrate the non-oncogenic and inactivated sites of the HPV antigen within the sequence, as well as the insertion positions of the HPV antigen into gD.
2. Including E2 and E5 in the experiment vaccine may improve the immunogenicity and the efficacy. However, in high-risk HPV types, it possesses highly conserved sites of E5 (DOI: 10.1002/jmv.27829, DOI: 10.1016/j.virol.2023.109946). Were the conservative sites of E5 knocked out during the design phase? Is it safe to use its full sequence without further investigation?
3. Strongly recommend a detailed review and meticulous revision of the entire manuscript and Figures, including addressing issues such as the explanation of the images is confusing (such as Fig. 3C and Fig. 3D), superscripts not properly formatted (such as "1010vp, ~ 1.5-2cm3, 5 ´ 105 TC-1 "), the figures quality is relatively low. These issues are significant and occur frequently throughout the manuscript.
4. Lines 387-393: Based on the experimental results, it is surprising to find that the antitumor efficacy of the E765 or E7652 antigen sequences carried by adenovirus, when not fused with gD, is comparable to the control group. This observation is consistent with similar findings in previous studies (2008 Nature Medicine, 2010 Clinical and Vaccine Immunology), regardless of whether DNA plasmids or adenovirus vectors were used. Both standalone pE7E6E5 or AdC6-E7652 exhibited outcomes similar to the control groups of pgD or AdC6-gD, suggesting that the effectiveness of the designed antigen sequences may be suboptimal. While the focus of the article seems to be on the remarkable tumor-assisting effect of gD, it is worth considering whether the ultimate efficacy of the vaccine design should be attributed to the effectiveness of the antigen sequences. Additionally, it would be beneficial to discuss whether the gene fusion of gD has been applied in clinical cases of tumor treatment and if so, how mature the approach is? If previous reports exist on its efficacy, they should be included in the discussion.
5. It is recommended to add a supplementary figure to illustrate the gating logic of flow cytometry for clearer presentation of the analytical approach.
Specific Comments
1. Lines 22-24: The HPV genome encodes 6 early proteins, lacking E3. For detailed information, it is recommended to refer to Reference 1 or the NCBI database.
2. Lines 25-27: It is suggested to refer to recent references or databases (https://www.hpvcenter.se/human_reference_clones/), as over 230 HPV types have been identified to date.
3. Line 145: “AdC vectors were diluted in sterile saline to 200 µl and injected at 1010vp i.m. into the hind legs of mice”. Previous studies administered adenovirus vector vaccines at doses of 2x107 IFU or 2x109 PFU (DOI: 10.1158/2326-6066.CIR-22-0174, DOI: 10.3390/ijms242015469), while lentivirus vector vaccines were administered at doses of 109 TU (DOI: 10.15252/emmm.202317723). Does the use of 1010 vp in this study raise concerns about dose-dependent safety issues?
4. Lines 206-207: Why is GA linker used in the design of E765, while AA linker is used in the design of E7652? Please explain the reason in the manuscript.
5. The figure legend for Fig. 1A (Lines 215-216, AdC68-gDE765, AdC68-gDE7652, and AdC6-gDE7652) does not match the figure caption (AdC68-gDE765, AdC6-gDE7652, and AdC6-gDE6752). "500bp" in Fig. 1B is obscured. In Fig. 2A, is "G+I-P+" mistakenly labeled as "G+I-+"? In Fig. 2A, the results of E6 and E7 peptide pool stimulation incorrectly write "AdC68-gDE765" as "AdC68-gDE76". In Fig. 3C, group labels are repeated. In Fig. 3E, both the figure caption and legend show "5x104 TC-1". Why is it described as "2x104 TC-1" in the results? Which is correct? In Fig. 3E, the letter "E" overlaps with the figure, indicating non-compliance with plotting standards. In Fig. 3G, the image is unclear. Is the injection quantity for TC-1 cells 106 or 104? In Fig. 5C, 5D, and 5F, "(mean±SEM" should be changed to "(mean±SEM)".
6. Line 281: Is it AdC6-E7652 or AdC6-gDE7652? The results describe AdC6-E7652, while the figure caption and legend show AdC6-gDE7652.
7. Lines 291-292: CD44 is typically associated with cell activation and memory. Are you detecting CD44+CD8+ double-positive cells to assess the activation level of CD8+ T cells and their involvement in the immune response (Fig. 2, Fig. 4)? Because in most studies, the usual approach is to detect the proportion of cytokine-secreting cells within CD3+CD8+ T cells (e.g., IFN-γ+CD8+/CD8+), including your previous research. Please provide the reason for this approach. Additionally, why are CD44 levels not measured in subsequent studies (Fig. 5)? Is it because the molecules you need to explore, such as CTLA-4, KLRG1, LAG3, PD-1, TIGIT, and TIM3, are all inhibitory molecules for T cell activation, so there's no need for CD44 to assess the activation level of CD8+ T cells?
8. Lines 351-364: The figure legends for Fig. 3C and Fig. 3D seem to be reversed.
9. Lines 373-378: Fig. 3J does not seem to be labeled in the figure.
10. Lines 405, 411: Is it Fig. 5E, Fig. 5F, or Fig. 3E, Fig. 3F?
11. Lines 399-400: The figures show AdC6-gDE7652 and AdC6-E7652. Why do lines 399-400 mention AdC68-gDE765 and AdC6-gDE7652?
12. Line 388: What are the TC-2 cells described? Are they TC-1 cells or TC-2 cells?
13. Lines 372, 374, 341, 349: Should there be spaces between "=" and letters or numbers, and between "1010" and "vp"? Should these be standardized?
14. Lines 429-430: It is recommended that the HPV16 whole genome-expressing C3 cell line (DOI: 10.1158/2326-6066.CIR-22-0174, DOI: 10.1002/(ISSN)1521-4141, DOI: 10.1080/2162402X.2019.1629259) be used for validation, or that the E2-expressing TC-1 cell line be constructed.
15. Line 511: In Fig. 5C and Fig. 5D, what is shown is "% marker-positive CD8+ cells", not "E7-positive CD8+ cells" as described in the figure legend.
Author Response
May 14th 2024
To the Editor,
We addressed the reviewers’ comments (in italic) as detailed below and hope the manuscript is now acceptable for publication. We fixed some additional typos and reduced the number of self-citations to 6
Sincerely
Ertl
Reviewer 1
Comments and Suggestions for Authors
The study developed two therapeutic vaccines targeting HPV-16 based on the AdC vector, one containing the full sequences of E5, E6, and E7, and the other containing different segments of E2, E5, E6, and E7. These vaccines, expressed by fusion to gD, demonstrated good immunogenicity and showed some efficacy in mice challenged with TC-1 cells. Particularly, the vaccine containing E2, when co-expressed with gD, enhanced, and broadened the specific CD8+ T cell responses to antigens. It not only elicited responses to E2 epitopes but also augmented responses to other viral oncoproteins such as E5 and E7. However, the mechanism by which E2 enhances immunity, and its efficacy remains unclear. This study has some innovative aspects, especially in the preliminary exploration of the anti-tumor effects of HPV early protein E2 and its potential mechanisms of action. My comments are as followed:
Major Comments
- Several experiment vaccines were constructed in present study. It is recommended to include a schematic diagram in the results section, clearly illustrating the amino acid sequences of each HPV antigen and the gD protein encoded by the adenovirus vector. This diagram should demonstrate the non-oncogenic and inactivated sites of the HPV antigen within the sequence, as well as the insertion positions of the HPV antigen into gD.
The requested diagram was added to Fig. 1.
- 2. Including E2 and E5 in the experiment vaccine may improve the immunogenicity and the efficacy. However, in high-risk HPV types, it possesses highly conserved sites of E5 (DOI: 10.1002/jmv.27829, DOI: 10.1016/j.virol.2023.109946). Were the conservative sites of E5 knocked out during the design phase? Is it safe to use its full sequence without further investigation?
We are not using full-length E5 but rather a fragment. The HA sequence present in the oncogenic types of HPVs is present in the E7652 sequence but not in the E765 sequence.
E5 sequences (wildtype, E765, E7652)
MTNLDTAYTTLLACFLLCFCVLLCVCLLIRPLLLSVSTYTSLILLVLLLWITAASAFRCFIVYIVFVYIP
CVLLCVCLLIRPLLLSVSTYTSLIILVLLLWITAASAFRCFIVYIAFVYIP
SVSTYTSLILLVLLLWITAASAFRCFIVYIVFVYIP
LFLIHTHARFLIT
LFLI
LFLIHTHARFLI
Considering the sequence is within a fusion protein the vast majority of which is secreted or transported to the cell surface within cells that in part due to the additional expression of the adenoviral antigens will get killed within 2-3 weeks by CD8+ T cells, we do not think that the presence of these two amino acids poses a risk. Nevertheless, we would like to thank the reviewers for pointing this out and will consider to remove these amino acids or conduct transformation assays before considering to test the vaccine in humans.
We included a sentence addressing this is the discussion: Sequences with E6 and E7 that may contribute to transformation were removed; a sequence within E5 that is only carried by oncogenic types of HPV (58) is present in the gDE7652 but not the gDE765 sequence and may have to be modified prior to their use in humans.
- 3. Strongly recommend a detailed review and meticulous revision of the entire manuscript and Figures, including addressing issues such as the explanation of the images is confusing (such as Fig. 3C and Fig. 3D), superscripts not properly formatted (such as "1010vp, ~ 1.5-2cm3, 5 ´ 105 TC-1 "), the figures quality is relatively low. These issues are significant and occur frequently throughout the manuscript.
We checked the manuscript carefully and changed the legends to Fig. 3C and D and formatting issues.
- Lines 387-393: Based on the experimental results, it is surprising to find that the antitumor efficacy of the E765 or E7652 antigen sequences carried by adenovirus, when not fused with gD, is comparable to the control group. This observation is consistent with similar findings in previous studies (2008 Nature Medicine, 2010 Clinical and Vaccine Immunology), regardless of whether DNA plasmids or adenovirus vectors were used. Both standalone pE7E6E5 or AdC6-E7652 exhibited outcomes similar to the control groups of pgD or AdC6-gD, suggesting that the effectiveness of the designed antigen sequences may be suboptimal. While the focus of the article seems to be on the remarkable tumor-assisting effect of gD, it is worth considering whether the ultimate efficacy of the vaccine design should be attributed to the effectiveness of the antigen sequences. Additionally, it would be beneficial to discuss whether the gene fusion of gD has been applied in clinical cases of tumor treatment and if so, how mature the approach is? If previous reports exist on its efficacy, they should be included in the discussion.
In mice the main CD8+ T cell response is directed against a single very immunodominant epitope within E7 that is present in all the vaccines. We agree without gD the vaccines only induce marginal responses. The selection of antigens may well be suboptimal especially for humans but nevertheless our results show very solid protection for the antigens expressed within gD. A vaccine expressing antigens of HBV is now in an ongoing phase I trial in patients with a chronic infection(https://www.prnewswire.com/news-releases/promising-first-in-human-phase-1b-clinical-study-data-from-vron-0200-a-novel-first-in-class-checkpoint-modifier-immunotherapy-for-chronic-hepatitis-b-virus-functional-cure-presented-as-late-breaker-at-2024-apasl-global-liver-me-302100640.html).
We included these two sentences into the beginning and end of the discussion: CD8+ T cell responses to the HPV antigen fragments were low or undetectable in mice immunized with the AdC vector expressing E7652 without gD, which may reflect that the selected sequences were suboptimal or more likely that HPV oncoproteins tend to be poorly immunogenic in mice.
Although clinical data of gD as a vaccine additive are not yet available its effects on human T cell responses is being explored in an ongoing clinical trials that is being conducted in patients with a chronic hepatitis B virus infection (https://clinicaltrials.gov/search?intr=VRON-0200).
- It is recommended to add a supplementary figure to illustrate the gating logic of flow cytometry for clearer presentation of the analytical approach.
This was added as a Suppl. Fig.
Specific Comments
Lines 22-24: The HPV genome encodes 6 early proteins, lacking E3. For detailed information, it is recommended to refer to Reference 1 or the NCBI database.
This was corrected.
Lines 25-27: It is suggested to refer to recent references or databases (https://www.hpvcenter.se/human_reference_clones/), as over 230 HPV types have been identified to date.
This was corrected.
Line 145: “AdC vectors were diluted in sterile saline to 200 µl and injected at 1010vp i.m. into the hind legs of mice”. Previous studies administered adenovirus vector vaccines at doses of 2x107 IFU or 2x109 PFU (DOI: 10.1158/2326-6066.CIR-22-0174, DOI: 10.3390/ijms242015469), while lentivirus vector vaccines were administered at doses of 109 TU (DOI: 10.15252/emmm.202317723). Does the use of 1010 vp in this study raise concerns about dose-dependent safety issues?
No. VP measures virus particles and is the FDA requested dosage unit while pfu measures infections units. In most of our preclinical adenovirus batches the ratio of vp to IFU exceeds 1:300 so that our doses are compatible with those of others.
Lines 206-207: Why is GA linker used in the design of E765, while AA linker is used in the design of E7652? Please explain the reason in the manuscript.
There was no particular reason – both are non-polar neutral amino acids so in this way they are similar. We added this sentence to the material and methods section: As both AAs are non-polar and neutral amino acids and thereby very similar this change should not have affected the structure of biological function of the fusion proteins.
The figure legend for Fig. 1A (Lines 215-216, AdC68-gDE765, AdC68-gDE7652, and AdC6-gDE7652) does not match the figure caption (AdC68-gDE765, AdC6-gDE7652, and AdC6-gDE6752). "500bp" in Fig. 1B is obscured.
Fig. 1A shows a Western blot with an anti-gD antibody. There was a typo in the capitation (AdsC6), which was corrected. 1B also shows a protein gel so there is no 500bp. We change the position of 500 bp in Fig. 1D.
In Fig. 2A, is "G+I-P+" mistakenly labeled as "G+I-+"?
Yes, this was corrected.
In Fig. 2A, the results of E6 and E7 peptide pool stimulation incorrectly write "AdC68-gDE765" as "AdC68-gDE76".
This was corrected.
In Fig. 3C, group labels are repeated.
This was corrected.
In Fig. 3E, both the figure caption and legend show "5x104 TC-1". Why is it described as "2x104 TC-1" in the results? Which is correct?
The cell doses shown in the figures are correct. The text was corrected accordingly.
In Fig. 3E, the letter "E" overlaps with the figure, indicating non-compliance with plotting standards.
This was corrected.
In Fig. 3G, the image is unclear. Is the injection quantity for TC-1 cells 106 or 104? In Fig. 5C, 5D, and 5F, "(mean±SEM" should be changed to "(mean±SEM)".
We changed : tumor size (mm3, mean ± SEM) to tumor size in mm3 (mean ± SEM).
- Line 281: Is it AdC6-E7652 or AdC6-gDE7652? The results describe AdC6-E7652, while the figure caption and legend show AdC6-gDE7652.
AdC6-gDE7652 is correct.
- Lines 291-292: CD44 is typically associated with cell activation and memory. Are you detecting CD44+CD8+ double-positive cells to assess the activation level of CD8+ T cells and their involvement in the immune response (Fig. 2, Fig. 4)? Because in most studies, the usual approach is to detect the proportion of cytokine-secreting cells within CD3+CD8+ T cells (e.g., IFN-γ+CD8+/CD8+), including your previous research. Please provide the reason for this approach. Additionally, why are CD44 levels not measured in subsequent studies (Fig. 5)? Is it because the molecules you need to explore, such as CTLA-4, KLRG1, LAG3, PD-1, TIGIT, and TIM3, are all inhibitory molecules for T cell activation, so there's no need for CD44 to assess the activation level of CD8+ T cells?
Most cytokine producing CD8+ T cells or T cells that stain positive with a tetramer or dextramer are CD44+. As only a fraction of CD8+ T cells are positive for CD44 a gate that includes CD44 increases the frequency of responding CD8+ T cells. For experiments where we look at low frequencies such as those testing responses to individual peptides or multiple cytokines, we prefer to use the CDD44+CD8+ gate and to be consistent we used this gate throughout the figures. Using the more typical CD8 gate would give as proportionally similar data and not change any of the conclusions.
- Lines 351-364: The figure legends for Fig. 3C and Fig. 3D seem to be reversed.
This was corrected.
- Lines 373-378: Fig. 3J does not seem to be abelled in the figure.
The H for the T cell assay was missing, and the J reflects the histology graphs. This was corrected.
- Lines 405, 411: Is it Fig. 5E, Fig. 5F, or Fig. 3E, Fig. 3F?
This was corrected in both cases to 3.
- Lines 399-400: The figures show AdC6-gDE7652 and AdC6-E7652. Why do lines 399-400 mention AdC68-gDE765 and AdC6-gDE7652?
Figure 4 compares responses to AdC68-gDE765 and AdC6-gDE7652, Figure 5 compares responses to AdC6-gDE7652 and AdC6-E7652. The text the reviewer refers to is linked to Figure 4
- Line 388: What are the TC-2 cells described? Are they TC-1 cells or TC-2 cells?
Typo. Should have been TC-1 and was corrected.
- Lines 372, 374, 341, 349: Should there be spaces between "=" and letters or numbers, and between "1010" and "vp"? Should these be standardized?
To be consistent, we added spaces.
- Lines 429-430: It is recommended that the HPV16 whole genome-expressing C3 cell line (DOI: 10.1158/2326-6066.CIR-22-0174, DOI: 10.1002/(ISSN)1521-4141, DOI: 10.1080/2162402X.2019.1629259) be used for validation, or that the E2-expressing TC-1 cell line be constructed.
Sorry but we will not be able to repeat all the challenge studies with an additional cell line. The interesting point of the manuscript is that E2 enhances T cell responses to other antigens and increases protection against a tumor that only expresses E7 and E6.
- Line 511: In Fig. 5C and Fig. 5D, what is shown is "% marker-positive CD8+ cells", not "E7-positive CD8+ cells" as described in the figure legend.
The text reads: [C,D] Percentages of E7 dextramer+CD8+ cells from spleen [C] or tumors [D] expressing the indicated markers. Which means the percentage of marker+ cells within the subset of T cells positive for the E7 tetramer.

Reviewer 2 Report
Comments and Suggestions for Authors
The manuscript "Pre-clinical immunogenicity and efficacy studies for therapeutic vaccines to human papilloma virus type 16-associated cancer" is very interesting. It relays on pre-clinical studies for therapeutic vaccines to human papilloma virus (HPV)-16 antigens (E5, E6, and E7 with or without E2 )combined with herpes simplex virus (HSV)-1 glycoprotein D (gD). The fusion antigens were expressed by chimpanzee adenovirus (AdC) vectors. Vaccines expressing either insert were highly immunogenic and showed efficacy in mice.
I consider this study prommicing having in mind the fact the patogenicity of HSV.
The manuscript introduction provides sufficient background and include all relevant references.
Material and methods are fine and the result support the conclusions. The cited references are relevant for research.
I agree with publication.
Congratulation.
Author Response
Reviewer 2:
Comments and Suggestions for Authors
The manuscript "Pre-clinical immunogenicity and efficacy studies for therapeutic vaccines to human papilloma virus type 16-associated cancer" is very interesting. It relays on pre-clinical studies for therapeutic vaccines to human papilloma virus (HPV)-16 antigens (E5, E6, and E7 with or without E2 )combined with herpes simplex virus (HSV)-1 glycoprotein D (gD). The fusion antigens were expressed by chimpanzee adenovirus (AdC) vectors. Vaccines expressing either insert were highly immunogenic and showed efficacy in mice.
I consider this study prommicing having in mind the fact the patogenicity of HSV.
The manuscript introduction provides sufficient background and include all relevant references.
Material and methods are fine and the result support the conclusions. The cited references are relevant for research.
I agree with publication.
Congratulation.
Comment from the authors: Thank you.

Reviewer 3 Report
Comments and Suggestions for Authors
In the manuscript "Preclinical studies of immunogenicity and efficacy of therapeutic vaccines against cancer associated with human papillomavirus type 16," the authors investigated the possible therapeutic effect of the vaccine targeting human papillomavirus (HPV)-16, which expresses immunogenic parts of the early antigens E5, E6, and E7 with 9 or without E2 fused to the glycoprotein D (gD) of herpes simplex virus (HSV)-1. This is an interesting topic that contributes to knowledge in the area, but certain issues must be corrected.
1. The objective of the manuscript must be mentioned in the abstract and introduction. The authors must also mention the gap that the manuscript will fill within the current knowledge in the abstract and introduction. Moreover, the authors must mention the conclusion of the work.
2. The authors should update their references. For example, there are actually more than 200 types of HPV found so far. Cervical cancer is the fourth and not the second cause of death due to cancer in women worldwide.
3. The results section must be improved by adding an explanation at the beginning of their description, and each subsection must have concluded: for instance, the data above suggest… or the information concluded that.
4. Provide better images; the ones presented in the manuscript are very pixelated. Moreover, the legends on the figures cannot be read.
5. Define all the abbreviations presented in the text and do so in their first appearance, either in the abstract or the body of the manuscript. For example, HPV.
Comments on the Quality of English LanguageNo comments
Author Response
Reviewer 3:
Comments and Suggestions for Authors
In the manuscript "Preclinical studies of immunogenicity and efficacy of therapeutic vaccines against cancer associated with human papillomavirus type 16," the authors investigated the possible therapeutic effect of the vaccine targeting human papillomavirus (HPV)-16, which expresses immunogenic parts of the early antigens E5, E6, and E7 with 9 or without E2 fused to the glycoprotein D (gD) of herpes simplex virus (HSV)-1. This is an interesting topic that contributes to knowledge in the area, but certain issues must be corrected.
- The objective of the manuscript must be mentioned in the abstract and introduction. The authors must also mention the gap that the manuscript will fill within the current knowledge in the abstract and introduction. Moreover, the authors must mention the conclusion of the work.
We added the requested information to the abstract and the introduction.
- The authors should update their references. For example, there are actually more than 200 types of HPV found so far. Cervical cancer is the fourth and not the second cause of death due to cancer in women worldwide.
We changed reference 6 and the two incorrect statements.
- The results section must be improved by adding an explanation at the beginning of their description, and each subsection must have concluded: for instance, the data above suggest… or the information concluded that.
This was added to sections of the result section that lacked a conclusion or an introductory sentence.
- Provide better images; the ones presented in the manuscript are very pixelated. Moreover, the legends on the figures cannot be read.
We increased the font size. We will upload high resolution images.
- Define all the abbreviations presented in the text and do so in their first appearance, either in the abstract or the body of the manuscript. For example, HPV.
All abbreviations are now explained.
Comments on the Quality of English Language
No comments
